# Fourth Ageism: Real and Imaginary Old Age

Paul Higgs * and Chris Gilleard

Division of Psychiatry, Faculty of Brain Sciences, University College London, London W1T 7NF, UK; c.gilleard@ucl.ac.uk
* Correspondence: p.higgs@ucl.ac.uk

**Abstract:** This paper is concerned with the issue of ageism and its salience in current debates about the COVID-19 pandemic. In it, we address the question of how best to interpret the impact that the pandemic has had on the older population. While many feel angry at what they see as discriminatory lock-down practices confining older people to their homes, others are equally concerned by the failure of state responses to protect and preserve the health of older people, especially those receiving long-term care. This contrast in framing ageist responses to the pandemic, we suggest, arises from differing social representations of later life, reflecting the selective foregrounding of third versus fourth age imaginaries. Recognising the tension between social and biological parameters of ageing and its social categorisations, we suggest, may offer a more measured, as well as a less discriminatory, approach to addressing the selective use of chronological age as a line of demarcation within society.

**Keywords:** ageism; COVID-19; fourth age; nursing homes; third age

## 1. Introduction

In a paper on ageism published in 2020, we argued that the term ageism has become a concept that has been extended too far, and used so broadly that it fails to specify exactly what it is that is being discussed [1]. Ageism is applied to all sorts of circumstances and levels as a way of explaining nearly all the negative situations and consequences associated with old age. Such overgeneralised use risks leading to an over-determination of processes on the basis of a single opposition, so much so that ageism becomes the explanation for all that is untoward in the lives of older people. Whether presented as the articulation of a set of beliefs serving the interests of a particular group, or as representing a particular logic underlying an external structural process within society, this all-encompassing, essentialisation of ageism covers up too many theoretical gaps. Consequently, we argued, it fails to provide a useful analytical framework for understanding the diverse social space that older people occupy in society.

This diversity has emerged most notably in the last decades of the 20th century and the early decades of this one. The consequence has been a profound transformation in the social relations of later life [2,3]. From an ascribed terminal destination in the life course, later life has become a more actively constructed social space. No longer reduced to a category of need set apart from the relations of production, later life has provided a widening arena for the emergence of what we have called cultures of the third age. This has encouraged a rejection of this residual location, whether as a personal identity or as a social category [4,5]. Seeing age as representing "nothing but a number" has become a popular motif driving a more socially and culturally inclusive agenda [6,7]. It also serves as a resource for combatting negative images of agedness as the essence of older people At the same time, these third age cultural tropes have become a stimulus for distancing the 'young old' 'the not yet old' and the 'still middle-aged' from those whose lives seem closer to the stereotype of decline and dependency of a 'real' old age in which 'older people' have been wrapped up for so long [8].

**2. Old Age, Later Life, and Their Third and Fourth Age Frameworks**

There are many ways of attributing causality to the emergence of these distinctions in shaping later life. Whether it is the pervasiveness of a consumerist youth culture that valorises an 'ageless ageing', or the progressive destandardisation of the institutionalised life course creating this desire for a more flexible identity, is open to debate [9–11]. What is less debatable, however, is that the vicissitudes of physiological ageing, including age-related cognitive impairments requires some conceptual separation within the expanding social space of later life. Paul Baltes [12] and Peter Laslett [13] are two of the more notable gerontologists who have used the idea of a fourth age to define one such category, applied to those where the combination of chronology and chronic illness betokens a terminal phase in people's lives. Less explicitly, Matilda White Riley used the term 'oldest old' to mark off that growing segment of the older population aged over 85 whose health is deemed a key factor in their experience of ageing [14].

While these conceptualisations of a 'real' old age—whether framed as a fourth age or the oldest old—diverge on what are its most salient features, they share a common position in seeking some line of distinction in the diversity of later life. Baltes, Laslett and Riley, each in their different ways, selected chronological age-typically 80 or 85 years-on which to draw a qualitative divide between those who are and those who are not yet 'really' old [15]. Most recently, the Japanese Gerontological Society have distinguished between a 'pre-old age' (from 65 to 74 years of age.) and a 'real' old age, reached on or after an individual's 75th birthday [16]. Some writers contend that this distinction should not be based upon chronology at all. They suggest an emphasis upon gender (with a 'masculine' agentic third age contrasted with a 'feminine' passive fourth age [17]). Others have argued that the distinction represents a switch in register. In this framework, the cultural, economic and social dimensions of life in the third age are contrasted to the corporeal, pathological experiences of the fourth age [18].

In our own work on the fourth age, we have argued for a conceptual dichotomy between the third and the fourth ages [19]. Each position necessarily leans upon the other, but less because of numerical ascription and more because of the cultural imaginary through which each is framed. The cultures of the third age operate against the shadows of a social imaginary of the fourth. Fears of an unwanted old age defined by physical frailty, immobility, and the diminution of agency serve as key motivating forces for third age consumerism. The fourth age is defined less by what it actually is than by what it is not. Its imaginary is shaped through its antithetical projection of a dependent old age and not the youthful, vital, healthy and successful ageing that feature so much in the range of books and magazines promoting third age lifestyles. Rather than the body being a site of performance, the body in the fourth age is one conjuring up pathos. It is within this tension between the opposing cultural frames of ageing that the idea of ageism needs situating. The more that later life is represented as an arena of lifestyle choice and fulfilment, the more age related frailty and disability become distanced from it, and become part of the "eternally aged other" [20] (p. 64).

It is unsurprising then, that the nursing home and assisted living facilities become the condensed image of this rejected old age. They represent a fate to be resisted, if not avoided altogether, a fate worse than death [21]. This fear cannot be reduced to the over determining ideology of ageism [22], nor to the cultural product of a 'malignant social psychology' permeating the long-term care system; one that undervalues the personhood of mentally and physically frail individuals [23]. Avoiding the 'natural' association of chronological agedness with illness and impairment is a salient feature of the cultures of the third age and its general resistance to decline narratives. The emphasis on health, leisure and self-actualisation flows naturally from currents already present within consumer society. The third age, in short, is directed toward a different set of outcomes, furthering the desire for a clear distinction from those seen to be displaying the markers of the fourth age. In contrast to the embodied freedoms and leisure by which third age lifestyles are promoted, in the fourth age we are presented with the disembodied images of physical

aids such as strollers and walking frames, or various forms of 'granny' wear or old-age products such as hearing aids, incontinence pads and dressing sticks.

The antipathy towards nursing homes as sites epitomising the fourth age extends to geriatric medicine as a whole, as well as other initiatives addressing the 'needs' and 'risks' imputed to older adults such as domiciliary services, meals on wheels and senior citizen centres. In a similar fashion to the subtle distinctions present in youth sub-cultures [24], these distinctions pervade the social space of later life. They now extend to the various public health responses to Covid-19 to which we now turn.

### 3. Third and Fourth Age Responses to COVID 19

There has been a relatively consistent worldwide response to the COVID-19 pandemic [25]. Lockdowns, quarantines, physical and social distancing, the wearing of masks and increased hand washing as well as sanitisation, are all policies that have been adopted by a variety of countries. This has resulted in a multiplicity of reactions from many differing political positions. Ageing and old age have not been immune to these fault lines. The inclusion of older people, along with people with various 'underlying' health conditions in lists of those needing to be shielded (and by implication kept apart from the rest of society) has provoked anger among many older age groups, incensed by these attributions of frailty. The use of chronological demarcations between those in the 'normal' population and those deemed automatically vulnerable has fuelled accusations of ageism [26]. If age is really just a number, however, what injustice is being perpetrated by selecting age as a cut-off used to place older people in a category as no different to the sick and infirm? There are many individuals aged over 70, they claim, whose fitness and flourishing is on a par with, or indeed may be somewhat better than the health of some of those in younger cohorts. Why should we, they ask, have our liberties curtailed by enforced 'shielding' on the dubious grounds of age [27]?

Viewing chronologically categorised older people en bloc as physically vulnerable, challenges the post-work identities that prior to the pandemic had been treated as relatively unproblematic. Such policies appear to be undermining the distinctions that have become so salient between a third and a fourth age. The high death rates of older people in nursing homes, in particular, has been one of the most noteworthy internationally reported features of the pandemic. In what seems to be a confluence of abjection [28], in country after country, nursing home residents have succumbed to the virus because of policies that did not prioritise their lives and effectively put them at greater risk [29]. The reasons vary from one government to another. In the UK, the fear of the NHS being overwhelmed led to older hospital patients being discharged from hospitals to nursing homes without first being tested for the virus [30]. In Sweden, a focus on giving citizens personal responsibility for taking precautions rather than implementing a mandatory lockdown contributed to Covid 19 coming into facilities for older people via the vector of care workers mingling with the population at large [31]. In Spain, the spread of the pandemic was such that some nursing home residents were abandoned by their fearful carers. Later these residents were to be found dead by army units sent to discover what had happened to them [32].

All these examples show how the impact of the pandemic was considerably worse for those enveloped within the institutions marked by the fourth age. Here ageism was undoubtedly occurring but it was a very specific form of ageism. There was an implicit assumption that this group (the care home population) constituted a less important category for policymakers than other groups when decisions about their needs were being made[1]. Their lack of significance was often underpinned by arguments that their deaths were inevitable or were a distraction from fighting for the lives of those more needing of and more likely to benefit from attention [33]. In the UK, some GP (family doctor) services had policies to issue 'Do Not Attempt Resuscitation' (DNAR) forms to their older and

---

[1] It has not been only the oldest people in care homes who have been relatively 'unprotected'. Many younger adults with mental and physical disabilities have also been reported to have been unnecessarily exposed to the virus and unnecessarily neglected when ill [32].

vulnerable patients, irrespective of whether or not they had been requested [34]. The UK National Institute for Health and Care Excellence (NICE) initially advised against the treatment of those categorised as 'frail' in order to restrict demand for hospital beds. However, when this was potentially also applied to younger, rather than just older, patients it was rapidly revised [35].

It is in the emergence of such practices where a more focused notion of '*fourth ageism*' has its use. The assumption that such lives need less consideration: as did the lives of those employed to look after them (in contrast to the heroism attributed to hospital staff). This ageism we term 'fourth ageism' because it is directed toward those who represent the unwanted, distasteful side of later life; in effect, those living under the shadow of the fourth age. As such, it is a very different matter from the issues connected to age discrimination that is often challenged at cultural, legal and political levels, not least by those rendered subject to such discrimination.

## 4. Symbolic Struggles and Social Spaces

The symbolic struggles over the place and position of later life in society have, we suggest, become more salient in recent years. Central to this development has been the emergence of third age cultures and third age lifestyles. Such cultures have become particularly significant for the consumer-driven economy of contemporary ageing societies. Within the context of this 'grey economy', and the refocusing of the state toward a greater emphasis upon citizens as consumers, it is predictable that 'old' ways of viewing 'old age' are seen as such: old-fashioned and outdated. Expressed with hardly any fervour a half century ago, the contemporary complaints over the 'invisibility' of older people reflect, if not their visibility, at least their voice, in opposing the oppression of non-recognition. In 2020 the worldwide media coverage of two men in their seventies battling for a position of immense power in one of the most powerful countries in the world is testimony to the visibility of agedness, or at least the continuing visibility of (some) older people. The presence of 'age' can be rendered invisible for reasons other than that of marginality. Among leaders of nation states, owners of property empires, the 'tycoons' of industrial capital, and the literary and artistic establishment, chronological age slips easily under the radar in comparison with other more socially salient characteristics- such as their wealth, power and celebrity status. In the fields of the third age, both in its objective and symbolic formation, the assets and resources attached to cultural, financial and social capital matter more than 'mere' chronology or corporeality.

As regards the fourth age, the reverse is more often the case. There are few 'symbolic' struggles over its meaning, status and value. Rather, such struggles as are evident, are those among those living and working alongside the most old, frail and infirm. They largely focus upon definitional entry to state controlled resources for long-term care, whether at home or in an institutional setting. The fourth age is neither a cultural field developed by the active practices of those assigned to its settings, nor is it a socio-cultural space where choice, distinction and self-expression are exercised. What social agency is realised is that of others, those neither aged, nor frail: the clinicians, family members and social care workers. Members of these groups determine the rules, establish the 'dividing practices' and frame what is to be done, albeit within a network of negotiations that notionally involve but which are never determined in the last instance by those citizens at most risk of being placed under the aegis of the fourth age.

## 5. Ageism: Real and Imaginary Old Age

Ignoring the significance of this divide between the third and fourth age causes much critical traction to be lost in combatting discrimination and marginalisation in later life. Applied without precision or focus, ageism becomes a more diffuse concept and risks striking a conspiratorial note that sees ageism behind everything that is in any way age-related. As we noted in our earlier paper on the ideology of ageism, the theoretical confusion regarding the causality of ageism leads to its over-extension, placing phenomena

under the same framework that reflect quite different trajectories, and which are embedded in different structures and are realised through very different lifestyles [1]. The neglect and isolation of residents in nursing homes is a tragedy not visited upon them purely because of their corporeality; there are many equally frail persons living in their own homes or with their families. It arises not least because of their invisibility as fellow citizens, but because they are counted less than those who are living 'freely' in the community, and because they have fewer people to count on. Such neglect, such marginality, can arguably be treated as the consequences of a 'fourth ageism'. This is where society and its institutions, avoids what Baltes called the 'darker side of ageing', and consequently fails to recognise residents of care homes and nursing homes as equal members of the public whose health is meant to be protected by the state, on behalf of the whole community.

The widespread anger felt by many older people toward the restrictions imposed by public health authorities in the pandemic, reflects a different matter: an equal mix of reality and imaginary whereby age is considered as 'nothing but a number'. On the one hand, it is important to stress that engagement with third age cultures is not contingent upon physical fitness and health status. A complex variety of individual, social and national factors play a significant part in creating the opportunities for and the space in which third age lifestyles can be realised through a different set of materialities than those attached to the fourth age. On the other hand, distinctions are important. Maintaining a sense of fitness-mental as much as physical-acts as a powerful motivating force in asserting the boundaries of the field and keeping the 'feared' form of old age at bay as is being angered by unthinking assumptions of decline and senescence.

At the same time, chronology also plays a part, both through the impact of social time (i.e., cohort and period effects) as well as through personal time (years spent ageing). As a social phenomenon, the internalised 'ageism' evident in the generational advocacy of an 'ageless ageing' is of a different character from that associated with the judged indignities and abjection of dependency and infirmity. While the former may be considered to reflect at most some kind of 'bad faith', or even 'inauthenticity', such tropes are considerably less restricting and life shortening than that other, more pernicious prejudice which both fears and forms the imaginary of the fourth age and its chronological countdown. This latter feature of fourth ageism does both. It mystifies infirmity while undermining moral status and human dignity. Whether or not a person dyes his or her hair, uses anti-ageing cream, receives periodic Botox injections, or is flattered by being judged 'young for their age' does not diminish his or her status, nor shorten his or her life. Middle-class, mid-life misgivings over age and ageing have become one part of the symbolic struggles of which contemporary life is constructed. Such imaginings are active, agentic and often necessarily adversarial; they involve participation through performance and consumption, as well as discourses and practices designed to assert distinction. This is quite unlike the social imaginary of the fourth age, which we would argue lacks both agency and contested subjectivity.

Here the realisation of such imaginings-of impairment and infirmity, of abjection agitation and suffering-arise in the discourses and practices that characterise the institutions and the practices of health and long-term, social care. These include crucially the conditions of labour characterising those care settings[2]. Social policies directed toward enlightening, improving, or reconstructing those institutions that operate under the *aegis* of the fourth age are however both imaginable and realisable. One of the legacies of the Covid 19 pandemic may be to expose and unveil the fourth ageism present in societies' arrangements for social care and thereby "give us the impetus to provide some more meaningful, lasting, and credible solutions to the funding and provision of social care" [36]. It is such realisations that may help fashion a fairer and more inclusive approach to all forms of long-term care, long after the over-seventies are back on the street, seeking to make themselves, not their age, visible players in society.

---

[2]　Of course similar criticisms could be made of many aspects of health care. Unlike social care, however, health care has long sought to avoid too close an encounter with age, as the history of 'geriatric medicine' well illustrates.

**Author Contributions:** P.H., C.G. were equally responsible for all aspects of the paper. All authors have read and agreed to the published version of the manuscript.

**Funding:** No funding was provided.

**Institutional Review Board Statement:** Not applicable.

**Informed Consent Statement:** Not applicable.

**Data Availability Statement:** Not applicable.

**Conflicts of Interest:** The authors declare no conflict of interest.

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
