# Peer review of "Fourth Ageism: Real and Imaginary Old Age"

_societies, doi:10.3390/soc11010012_

Round 1
Reviewer 1 Report
Thank you for submitting such an interesting manuscript. This paper greatly contributes to age studies and the concept of the Fourth Age. It also provides valuable insights into ageism and current debates about the Covid-19 pandemic and its consequences.
Author Response
Thank you for your positive comments on our paper. We have made some adjustments to make the paper clearer and the sentences shorter.
Reviewer 2 Report
Review of the manuscript:
Fourth Ageism: real and imaginary old age
Summary
This article aims to discuss the debates about the impact of Covid-19 pandemic on older population and the disparity of the judgements towards this; Suggesting considering
social and biological parameters of ageing to offer a less discriminative approach compared to the use of chronological age as a line of delineation. However, it is poorly addressed how the parameters mentioned in the abstract could offer a better approach towards interpretation of the pandemic effects on the older population.
The introduction needs to be improved. Also, please avoid using very long sentences throughout the whole paper. It is not very clear to the reader what every part is suggesting.
Additional explanation is needed for the information given in the introduction, and also the current knowledge around “third” and “fourth” age.
It is not defined how some express anger towards the discriminatory lock-down practices confining older people to their homes or towards how properly the elderly health is preserved
The conclusion needs further work, it restates what has been remark in the introduction, but does not provide insights. Authors should improve the conclusion creating a good flow for the reader
Also, there is structural flaws hindering the manuscript from giving a clear message to the readers. Some sentences are so confusing.
What is concerning, is that different parts of the paper seem to be not relevant or not well organized so that the readers would understand the flow of the manuscript.
There is not enough clear headings which could be confusing.
Author Response
Thank you for your comments we have tried to make our argument clearer and have shortened some of the longer sentences to improve clarity of argument
Reviewer 3 Report
This manuscript addressed the authors concerned with the issue of ageism and its salience in current debates about 8 the COVID-19 pandemic.
This article is likely a Short Communication, if so, I have only suggest to complete the lines 4-7 to show the real affiliations and e-mails.
I think it can be accepted with minor revision (to complete authors’ data on Front Matter). I have no more comments.
Author Response
We have shortened some of the sentences to improve clarity of expression as requested
Round 2
Reviewer 2 Report
The titles are better organized and and the argument seems to be clearer to the reader. Still needs some minor changes such as coherency in the discussion is needed.